# Deformation Behavior of Nanocrystalline Body-Centered Cubic Iron with Segregated, Foreign Interstitial: A Molecular Dynamics Study

**DOI:** 10.3390/ma13235351

**Published:** 2020-11-25

**Authors:** Ahmed Tamer AlMotasem, Matthias Posselt, Tomas Polcar

**Affiliations:** 1Department of Mechanical Engineering, Faculty of Engineering and Physical Sciences, University of Southampton, Southampton SO17 1BJ, UK; T.polcar@soton.ac.uk; 2Department of Physics, Faculty of science, Assiut University, Assiut 7156, Egypt; 3Helmholtz-Zentrum Dresden—Rossendorf, Institute of Ion Beam Physics and Materials Research, Bautzner Landstraße 400, 01328 Dresden, Germany; m.posselt@hzdr.de

**Keywords:** iron, molecular dynamics, segregation, dislocation, twinning

## Abstract

In the present work, modified embedded atom potential and large-scale molecular dynamics’ simulations were used to explore the effect of grain boundary (GB) segregated foreign interstitials on the deformation behavior of nanocrystalline (nc) iron. As a case study, carbon and nitrogen (about 2.5 at.%) were added to (nc) iron. The tensile test results showed that, at the onset of plasticity, grain boundary sliding mediated was dominated, whereas both dislocations and twinning were prevailing deformation mechanisms at high strain. Adding C/N into GBs reduces the free excess volume and consequently increases resistance to GB sliding. In agreement with experiments, the flow stress increased due to the presence of carbon or nitrogen and carbon had the stronger impact. Additionally, the simulation results revealed that GB reduction and suppressing GBs’ dislocation were the primary cause for GB strengthening. Moreover, we also found that the stress required for both intragranular dislocation and twinning nucleation were strongly dependent on the solute type.

## 1. Introduction

Nanocrystalline (nc) metals’ own superior mechanical properties make them great candidates for high-strength applications. In literature, several experimental and theoretical studies on mechanical properties of nc metals reveal that strengthening could be achieved not only by controlling their grain size but also by tailoring and doping their grain boundary [1,2,3,4,5,6,7]. It was reported that the deformation behavior and fracture of nanocrystalline materials are significantly driven by grain boundaries (GBs) and their underlying structure [8,9]. For instance, it was reported that alloying and GB segregation of foreign atoms can improve the strength of nc metals due to preventing grain growth, as evidenced by the enhanced thermal stability of a number of binary nanocrystalline alloys in comparison to their single-component counterparts [10,11,12,13,14,15]. In the case of strengthening of nc metals by GB segregation, the size and type of the foreign atoms, the available molar fraction of GB, and relaxation of GB are vital parameters [16]. For example, nitrogen and carbon are added to steels to improve their strength. However, experiments have shown that segregated carbon has a more strengthening effect than nitrogen. It was found that alloying of ferrite by carbon leads to an increase of their yield strength and the Hall–Petch (HP) coefficient increased greatly from 100 to 600 MPa·μm^−1/2^ In contrast, the HP coefficient was barely influenced by adding nitrogen [17,18,19]. In an attempt to explain the differences between the role of carbon and nitrogen on the HP coefficient, the results of atom probe tomography (APT) by Takahashi et al. [20] demonstrate that both nitrogen and carbon have a similar tendency to segregate at the boundary of ferritic steel. However, coefficient increment per unit interfacial excess for carbon is higher, which may explain why carbon has a significant influence on the ferrite strength. On the other hand, theoretical first principles’ results by Wu et al. [21] reveal that carbon acts as cohesion enhancer in body-centered cubic (bcc)Fe grain boundary while nitrogen weakens GBs [22].

In other work by Takuro et al. [23], they studied the deformation mechanisms of austenite steels, observed by electron backscatter diffraction (EBSD). Their results revealed the differences in deformation microstructure between carbon/nitrogen-added austenitic steels. They reported that adding carbon enhances deformation by twinning and ε-martensite phase transformation. In the case of nitrogen-added austenitic steels, the propensity to deform via dislocation was observed. The competition among deformation mechanisms was explained in terms of the variation of stacking fault energy (SFE) with the type of solute atoms.

Despite extensive experiments that strongly indicate that carbon and nitrogen segregation to GBs can improve the strength of nc ferrite, the roles of carbon and nitrogen in the strengthening and plasticity of these materials are still not fully understood. In the present work we utilized the large-scale atomistic simulations with classical interatomic potentials for Fe-C and Fe-N to investigate the effect of carbon and nitrogen on the deformation mechanisms of ferritic steel. Furthermore, the impact of carbon and nitrogen on the cohesive energy, atomic volume, and stacking faults was investigated. Such investigations will improve the knowledge on the mechanical properties of the nc ferritic steels and will contribute to widening the applications of these materials.

## 2. Simulation Details

A nanocrystalline bcc iron sample, depicted in Figure 1, was constructed following the Voronoi tessellation method [24], implemented within Atomsk tool [25]. In this method, a sample with average grain size, *D* = 14.12 nm, was built by filling in a cubic box with randomly distributed seeds. Then bcc lattice grains with random orientations were generated from the seeds. The simulation box had size of 350 × 210 × 170 Å^3^ along the x, y, and z directions, respectively, containing about 1,100,000 atoms. To eliminate free surface effects, the simulation cell was periodic in all directions. In order to avoid high energy configuration, the overlapped atoms within 0.7 Å distances were removed, followed by structural minimization using the conjugate gradient method. Then, the system was thermally equilibrated to 300 K and 0 GPa pressure for 200 ps using isobaric/isothermal constant number of particles, constant pressure and constant temperature (NPT) ensemble, so that the total stress became zero. The time step of 1 fs was chosen for our simulations. The common neighbor analysis (CNA) [26] was performed to distinguish lattice and nonlattice atoms, i.e., GB atoms. It was found that about 15% of the total number of atoms were located at grain boundaries. Then, about 17.5% of the GB atoms were randomly replaced by either C or N atoms, which corresponded to 2.5% of all atoms in the sample. This concentration was much higher than the solubility of carbon/nitrogen in bcc-iron so that under equilibrium conditions the formation of precipitates of new phases such as cementite and graphite was expected [27]. In the present work, these effects were neglected, and the primary focus was on both GB strengthening and the deformation behavior of nanocrystalline bcc-iron with C/N GB segregants. It should be pointed out here that the present investigations can be considered as a case study, which led to qualitative results without claiming that all phenomena related to the influence of the solutes were considered. The reason why a relatively high solute concentration of C and N was chosen was mainly due to the expectation that in this manner the effect of the solutes would become clearer than at low concentrations. That means the focus was on the discussion of the qualitative behavior instead of on obtaining precise quantitative results. Carbon and nitrogen were randomly placed in the predefined GBs and only the solute type was changed, whereas the other properties, such as the grain boundary structure, the solute atoms distribution, and annealing, were fixed. This approach is justified since the experimental measurement by atom probe tomography (APT) [20] revealed that both nitrogen and carbon tend to segregate with a similar amount at GBs of ferritic steel.

During dynamic loading, the sample was subjected to uniaxial tensile load with constant strain rate, 5 × 10^8^ s^−1^, along x direction, whereas the pressure in both y and z directions was kept at 0. This strain rate was considerably high compared to the lab experiments; however, it was enough to capture qualitatively grain boundary activity as well as dislocations mediated by plastic deformation.

The interaction between Fe-C [28] and Fe-N [29] atoms is described by the second nearest neighbor modified embedded atom method (2NN-MEAM) potentials. The potentials’ parameters were developed by fitting to the experimental data of Fe-C and Fe-N dilute systems. These data included the dilute heat of solution, the location of interstitial atoms, the migration energy of solute atoms, the vacancy–carbon binding energy and its configuration in bcc and face-centered cubic (fcc) iron, and the enthalpy of formation and lattice parameters of nitride phases. Thus, the current potentials could be used reliably to simulate the interactions between carbon and nitrogen interstitials with dislocations and grain boundaries and to investigate the effects of carbon and nitrogen on various deformation and mechanical behaviors of bcc/fcc iron.

During tensile simulation, the identification of dislocations was made via the dislocation extraction algorithm (DXA) [30], while the structural defects were analyzed by the Crystal Analysis Tool (CAT), based on adaptive common-neighbor analysis, developed by Stukowski et al. [31,32]. Visualization of all molecular dynamics (MD) simulation snapshots were made via the open source software Ovito [33] and Atom Viewer [34,35].

## 3. Results and Discussion

Before applying the tensile deformation, the as-prepared samples underwent thermal annealing at 300 K for 200 ps. This procedure is important to allow the system to relax and remove undesired high-energy configurations. Here, we focused only on the role of the grain boundary segregation to the strength of nanocrystalline samples. Figure 2 illustrates the influence of segregated atoms on the potential energy per atom for different nc samples.

Obviously, the GB area is characterized by atoms having higher energies compared to grain interior atoms due to the lack of ordering of GB atoms. In order to quantitatively assess the effect of the segregated atoms on the material strength, one may estimate the GB energy, given by $E_{GB}=(E_{T}-E_{SC})/A_{GB}$, where $E_{T}$, $E_{SC}$ and $A_{GB}$ are the total energy of nc sample, the total energy of its corresponding single crystal, and the total area of GB, respectively [16,36]. However, in MD simulations, $A_{GB}$ is not easily determined. Therefore, we developed another technique based on the energy per GB atoms. In this method, we first identified GB atoms using CNA technique then GB atoms were classified according to their energy per atom ($E_{at}^{GB}$) with respect to the cohesive energy of bcc iron atom ($E_{coh}^{Fe}$) in perfect lattice (−4.29 eV/atom, according to the current interatomic potential). Then, the ratio, $R(E_{at}^{GB})$, between the number of GB atoms with $\left(E_{at}^{GB}>E_{coh}^{Fe}\right)$ to those having $\left(E_{at}^{GB}<E_{coh}^{Fe}\right)$ was determined. Figure 3 displays the value of $R(E_{at}^{GB})$ corresponding to each nc sample. As evident from Figure 3, the value of $R(E_{at}^{GB})$ was the largest for the pure nc-Fe sample. This was already obvious from Figure 2, since the number of GB atoms with energy greater than that in the grain was very high. We also found that adding nitrogen atoms to GB led to a slight decrease of the value of $R(E_{at}^{GB})$, whereas adding carbon atoms caused a drastic increase of the number of GB atoms having $\left(E_{at}^{GB}<E_{coh}^{Fe}\right)$. This result suggests that carbon has a more significant impact, compared to nitrogen, on improving ferrite strength by great reduction of GB energy. Our findings are in line with the results of density function theory (DFT), where it is reported that adding carbon and nitrogen improves the cohesion of GB in bcc-Fe [22,37].

Additionally, the results of DXA analysis, as depicted in Figure 4, revealed the absence of intragranular dislocations in all samples, and only grain boundary dislocations (GBDs) were observed in the case of pure nc-iron. Such GBDs, alongside GB triple junctions, act as sources of dislocations’ emission.

Figure 4 demonstrates that the segregated C and N suppress the formation of dislocations at GBs and, therefore, may enhance the strength of the GBs. The present results support the explanation given by Takaki [18] for the substantial increase in Hall–Petch coefficient $k_{y}$ for carbon-added ferritic steel compared to that for nitrogen-added ferritic steels. This phenomenon is in accordance with Cottrell’s model [38], in which it is assumed that the presence of interstitials at GB pin the dislocation emission site at grain boundary.

### 3.1. Stress–Strain Curves of nc Samples

Figure 5 shows the typical stress–strain curves of all nc samples uniaxially loaded up to 16% strain. Three different regimes can be highlighted: (1) elastic regime up to about 2% strain, (2) GB sliding regime, and (3) plastic regime. The 0.2% offset yield strength was 1.09, 1.28, and 1.12 GPa for Fe, Fe-C, and Fe-N nanocrystalline samples, respectively. Beyond this value, the stress increased with strain up to peak stress, corresponding to the emission of the first dislocation, at about 7.61, 7.82, and 7.46 GPa for nc Fe, nc Fe-C, and nc Fe-N samples, respectively. The peak stress was greatly amplified compared to the corresponding experimental values [17,18] because of the high strain rate inherent in MD simulation. Hence, the peak stress could not be used as a measure of material strength; rather, the flow stress ($\sigma_{\mathrm{flow}}$) was used since it is insensitive to the strain rate.

The value of $\sigma_{\mathrm{flow}}$ was obtained by averaging the total stress-over-strain interval of 14–16%. The values of $\sigma_{\mathrm{flow}}$ were 3.77, 4.34, and 4.08 GPa for Fe, Fe-C, and Fe-N nanocrystalline samples, respectively. In general, we found that both carbon and nitrogen tended to increase the material strength, as indicated by the increase of the flow stress of nc-iron. However, while carbon increased the value of $\sigma_{\mathrm{flow}}$ by about 15%, the $\sigma_{\mathrm{flow}}$ value was larger only by 8% when adding the same amount of nitrogen. These findings are in qualitative agreement with experimental observation available in [17,19]. It was shown that the Hall–Petch coefficient, $k_{y}$ and, subsequently, the yield strength of ferritic steels increased with the addition of carbon and nitrogen. Also, it has been reported that carbon atoms have greater influence on the ferritic strength compared to that of nitrogen atoms.

### 3.2. Effect of Carbon/Nitrogen Segregation on the Plastic Deformation Processes

In the following we shall investigate the influence of solute elements on the plasticity of nc samples by analyzing the deformation mechanisms at different strain values. As mentioned in Section 3.1, the stress–strain curves exhibited linear dependence in strain range up to 2%. In this regime no dislocations or twin activities were observed. Thus, it was assumed that the deformation mechanism in this stage was rather due to GB activities, i.e., the deformation was localized in GB. We examined this assumption by calculating the von Mises atomic local shear strain ($\eta_{\mathrm{vm}}$), given by Equation (1) [26].
(1)$$\eta_{\mathrm{vm}}=\sqrt{\eta_{xy}^{2}+\eta_{yz}^{2}+\eta_{xz}^{2}+\frac{1}{6}\left[{\left(\eta_{xx}-\eta_{yy}\right)}^{2}+{\left(\eta_{yy}-\eta_{zz}\right)}^{2}+{\left(\eta_{xx}-\eta_{zz}\right)}^{2}\right]}$$

The spatial distribution of ($\eta_{\mathrm{vm}}$) of different nc samples at total strain 2% is shown in Figure 6. Obviously, the atomic strain was highly localized at GBs, compared to the grain interior atoms, suggesting that GB sliding was active in this regime. It was noticed that the largest atomic shear strain was observed for the nc iron and adding foreign atoms led to the decrease of the value of $\eta_{\mathrm{vm}}$.

Moreover, to examine the impact of solute atoms on the grain boundaries by calculating the atomic displacement perpendicular to the loading direction $u_{z}$. Figure 7 displays the probability density of $u_{z}$ per atom for the lattice and non-lattice atoms. As illustrated in Figure 7, the value of $u_{z}$ of non-lattice atoms decreased when foreign atoms were added to ferrite, i.e., an increase of GB sliding resistance, suggesting that the enhancement of the material strength was due to stabilizing the GB and preventing grain growth.

To understand the influence of foreign atoms on the GB sliding, we calculated the free volume of GB atoms. This quantity is particularly useful as stress-driven free volume migration is one of the mechanisms that mediates GB sliding [39]. We first identified GB atoms using CNA technique and then we applied Voronoi analysis, implemented in VORO++ [40], to calculate the volume per GB atom $\Omega_{at}^{GB}$. We defined $R(\Omega_{at}^{GB})$ as a ratio between the number of GB atoms with $\Omega_{at}^{GB}>\Omega_{at,0}^{Fe}$ to the number of GB atoms having $\Omega_{at}^{GB}<\Omega_{at,0}^{Fe}$ where $\Omega_{at,0}^{Fe}$ is the cohesive energy of Fe atom in bulk bcc lattice and equals 11.77 Å^3^/atom. Figure 8 shows the value of $R(\Omega_{at}^{GB})$ corresponding to different nc samples. Obviously, nc iron had the largest $R(\Omega_{at}^{GB})$, i.e., the largest free volume; thus, the propensity of GB sliding was higher. When foreign atoms such as carbon or nitrogen (with a smaller size compared to iron atoms) are added, the value of $R(\Omega_{at}^{GB})$ decreases. As a result, the available free volume that can assist GB sliding was reduced, leading to an increase of GB sliding resistance.

On the other hand, in the large strain regime, the plasticity of nc samples was mediated by additional two competing deformation mechanisms, intragranular dislocations and deformation twinning, as shown in Figure 9. In general, two kinds of dislocations, which are commonly observed in bcc-Fe, were found, namely, dislocations with burgers vectors 1/2<111> and <100> [41]. In order to quantify the influence of the solute type on the deformation mechanisms accompanied with the plasticity of nc iron, we calculated the variation of intragranular dislocation density with the applied strain. The intragranular dislocation was identified as follows. At first, we used CNA technique to distinguish both grain boundary and grain interior atoms. Then GB atoms were removed and, finally, the DXA was applied to extract the total number of dislocations, which was divided by total volume of all grains. Figure 10a shows the variation of intragranular dislocation density with applied strain. From Figure 10a, two important aspects can be highlighted: (1) The dislocation density was drastically reduced when solute atoms were added to GB and (2) an increase of the strain was required for the nucleation of intragranular dislocation for samples containing carbon and nitrogen, respectively. Thus, we assumed that both carbon and nitrogen play a key role in the nucleation of dislocations. 

Additionally, the twinning fraction observed during tensile test for nc samples was determined and is presented in Figure 10b. Similarly, we found that the strains at which twins nucleate increased as 0.055, 0.068, and 0.075 for nc Fe, nc Fe-N, and nc Fe-C, respectively. With further strain, deformation twin was more dominated over dislocation activity mechanisms in the case of nc samples containing solute atoms, as indicated by the increase of the twin fraction.

To understand the effect of segregation of foreign atoms on the deformation mechanisms of nc bcc-iron, the generalized planar fault energy (GPFE) of bcc-Fe single crystal with/without foreign atoms was studied (see Appendix A). The GPFE curve can accurately provide the values of the stable fault energy ($\gamma_{\mathrm{sf}}$), the unstable fault ($\gamma_{\mathrm{usf}}$), and the unstable twinning fault ($\gamma_{\mathrm{utf}}$) (see Table A1). These values are important to determine the competition among deformation mechanisms. Using the value of $\gamma_{usf}$, we determined the stress required for nucleation of twins based on the pole mechanism proposed by Cottrell and Bilby [42]. According to pole mechanism, the critical nucleation twinning stress ($\tau_{o}$) is given by $\tau_{o}=\frac{\gamma_{usf}}{b}$, where b is the dislocation Burgers vector. The calculated values of $\tau_{o}$ were 0.37, 0.71, and 1.31 GPa for Fe, Fe-N, and Fe-C nanocrystalline samples, respectively. Interestingly, the calculated value of $\tau_{o}$ for nc Fe was close to the typical experimental value of body-centered cubic iron (<0.5 GPa [43,44]). The increase of the twinning stress of ferrite when carbon is added to ferritic steel was revealed experimentally by Magee et al. [45].

On the other hand, we calculated the ratio, ($\gamma_{\mathrm{sf}}/\gamma_{\mathrm{usf}}$), related to the energy barrier for dislocation nucleation and ($\gamma_{\mathrm{utf}}/\gamma_{\mathrm{usf}}$), a measure of the energy barrier for twinning nucleation. We found that the lowest ($\gamma_{\mathrm{sf}}/\gamma_{\mathrm{usf}}$) and ($\gamma_{\mathrm{utf}}/\gamma_{\mathrm{usf}}$) values were obtained for bcc-Fe; hence, both dislocation and twinning were observed simultaneously at relatively small strains, as illustrated in Figure 9a–c. The ratio of $\gamma_{\mathrm{utf}}/\gamma_{\mathrm{usf}}$) increased from nc Fe via nc Fe-N to nc Fe-C, which resulted in an increase of the strain required for dislocation nucleation, as depicted in Figure 10a. Similarly, the ratio ($\gamma_{\mathrm{utf}}/\gamma_{\mathrm{usf}}$) increased, indicating that the strain required for twinning nucleation also increased, as shown in Figure 10b. Looking at Figure 9 and Figure 10, they clearly show that dislocation activity was reduced in nc Fe-N and nc Fe-C, whereas twins became more pronounced. The above results suggest that carbon and nitrogen play a significant role during the nucleation stage of twins. We found that carbon atoms promote the plastic deformation via twinning in ferrite steels. 

Finally, the formation of vacancies during plastic deformation was observed. Obviously, as depicted in Figure 9b and Figure 10c, the fraction of vacancies was more pronounced in the case nc iron sample. The detailed analysis of vacancies’ fraction with strain, cf. Figure 10c, indicates that the onset of vacancies’ formation occurred at higher strain for samples containing foreign atoms. Thus, the presence of carbon and nitrogen suppressed the formation of vacancy nanoclusters. Also, it was noticed that the strains at which the vacancy formation starts were comparably equal to the strains at which twins or dislocations nucleated. This fact suggests that the formation of vacancies was closely related to the interplay between dislocations and deformation twins during deformation. The formation of vacancy nanoclusters was not surprising, as they may be generated as a result either from a rough motion of a dislocation under high stress, Figure 9b, which eventually leads to the formation of lattice defects as reported in [46] or from the deformation twins, Figure 9e. This mechanism has been experimentally verified by electron microscopy of high-speed deformation of metals and later explained by Seeger [47].

## 4. Conclusions

In the present study we performed MD simulations to explore the role of grain boundary segregated carbon and nitrogen on the plasticity of nanocrystalline bcc-iron. Our MD results indicate that GB strengthening occurred because of two primary mechanisms, the reduction of GB energy and the pinning of the GBs’ dislocations. Moreover, the flow stress of nc bcc-iron increased by about 15% and 8% upon addition of C or N, respectively. During tensile deformation, the grain boundary sliding was dominant at early stages of plasticity, while at large strain both dislocation activity and deformation twinning contributed to deformation mechanisms. Adding carbon or nitrogen to bcc-iron also reduced the free atomic volume fraction and, consequently, the grain boundary sliding resistance increased. Additionally, the calculated critical nucleation stress for twins and dislocation was higher for samples with solute atoms compared to pure nc bcc-iron. The simulation results related to the generalized planar fault energy curve clearly showed that the values of stacking and twinning fault energies varied considerably with carbon and nitrogen content. 

## Figures and Tables

**Figure 1 materials-13-05351-f001:**
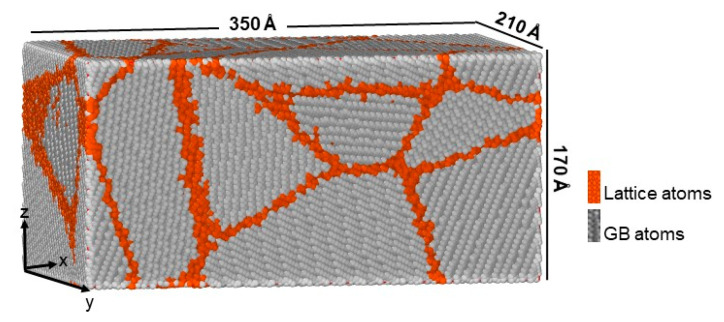
Simulation cell of nc ferrite with average grain size of 8 nm.

**Figure 2 materials-13-05351-f002:**
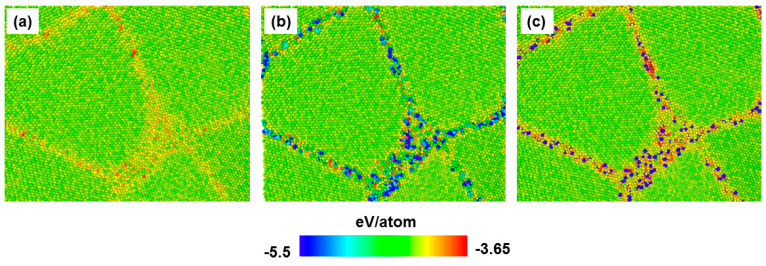
MD snapshot showing cross-sectional view of y-z plane of the potential energy per atom for annealed structures at 300 K. (**a**) Pure iron, (**b**) Fe-C, and (**c**) Fe-N. (For interpretation of the references to color in this figure, the reader is referred to the web version of this article).

**Figure 3 materials-13-05351-f003:**
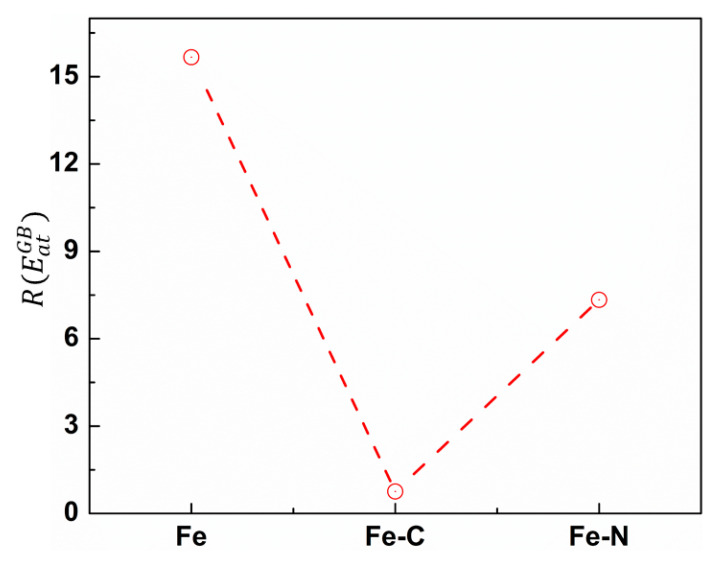
The ratio between the number of GB atoms with $\left(E_{at}^{GB}>E_{coh}^{Fe}\right)$ and those having $\left(E_{at}^{GB}<E_{coh}^{Fe}\right)$ corresponding to different nc samples.

**Figure 4 materials-13-05351-f004:**
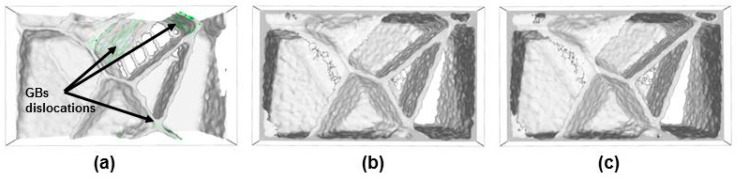
Nanocrystalline samples obtained after annealing at 300 K for 100 ps. (**a**) Fe, (**b**) Fe-C, and (**c**) Fe-N. Atoms are omitted for better visualization.

**Figure 5 materials-13-05351-f005:**
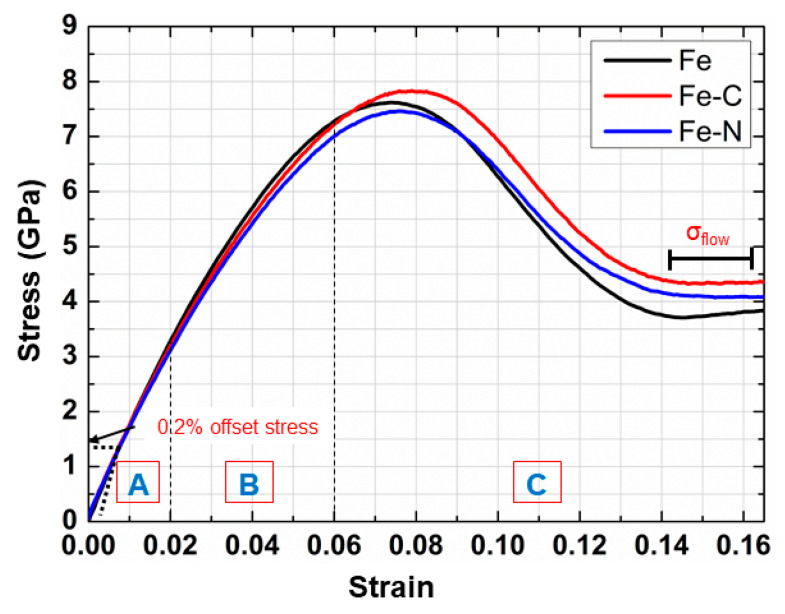
Typical stress–strain curves during tensile deformation of nc samples at strain rate 5 × 10^8^ s^−1^.

**Figure 6 materials-13-05351-f006:**
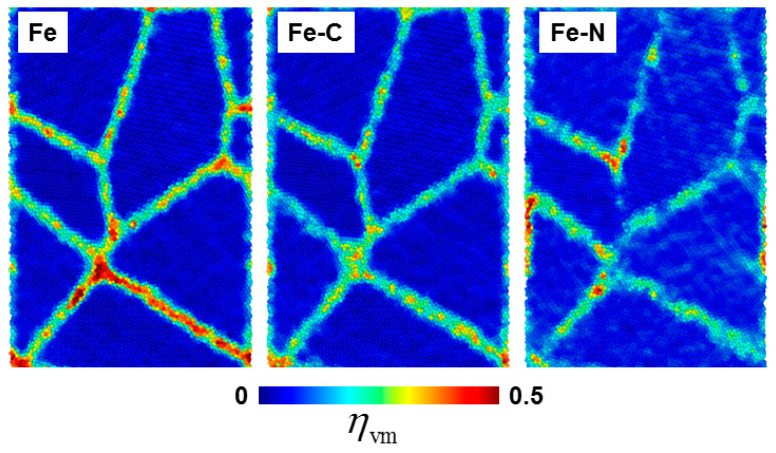
Sliced snapshots showing the spatial distribution of atomic-level strain ($\eta_{\mathrm{vm}}$) obtained at 2% strain for different nc samples. (For interpretation of the references to color in this figure, the reader is referred to the web version of this article).

**Figure 7 materials-13-05351-f007:**
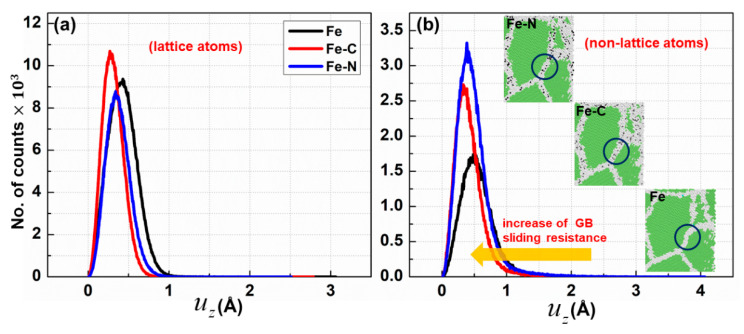
Distribution of atomic-level displacement ($u_{z}$) for different nc samples at 2% strain (**a**) grain interior atoms (lattice atoms), (**b**) grain boundary atoms (non-lattice). The circles denote GBs that were investigated.

**Figure 8 materials-13-05351-f008:**
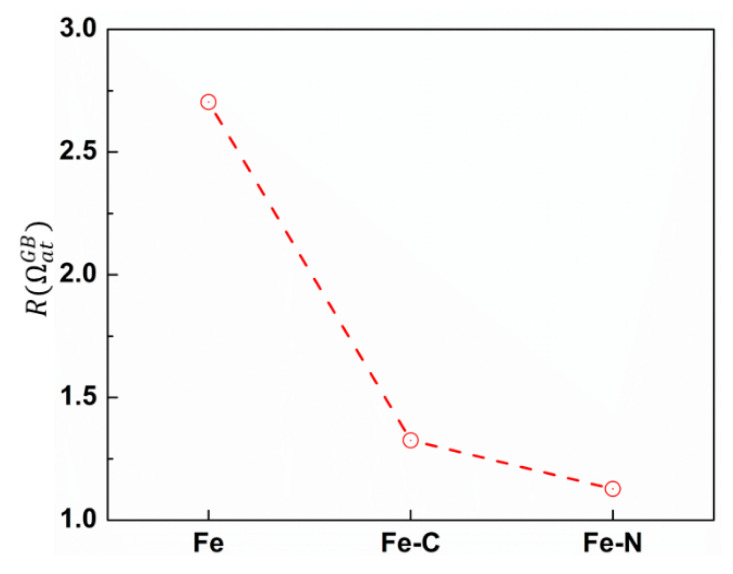
The ratio, $R(\Omega_{at}^{GB})$, between the number of GB atoms with $\Omega_{at}^{GB}>\Omega_{at,0}^{Fe}$ and the number of GB atoms having $\Omega_{at}^{GB}<\Omega_{at,0}^{Fe}$, corresponding to different nc samples.

**Figure 9 materials-13-05351-f009:**
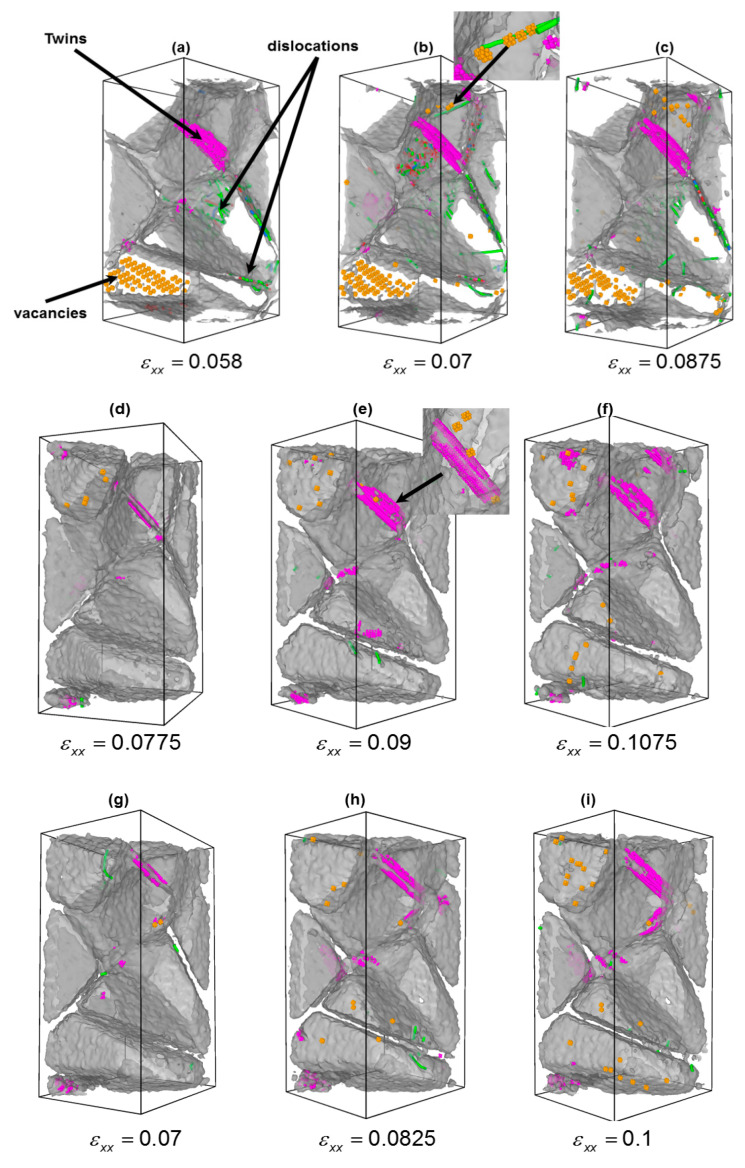
MD snapshots showing the evolution of various deformation mechanisms observed during tensile test of nc Fe (upper panel), nc-Fe-C (middle panel), and nc-Fe-N (lower panel) at different strains. (For interpretation of the references to color in this figure, the reader is referred to the web version of this article). green, purple and orange colors denote dislocation, twins and vacancy clusters, respectively. (**a**–**i**) represents the microstructural evolution under applied strain.

**Figure 10 materials-13-05351-f010:**
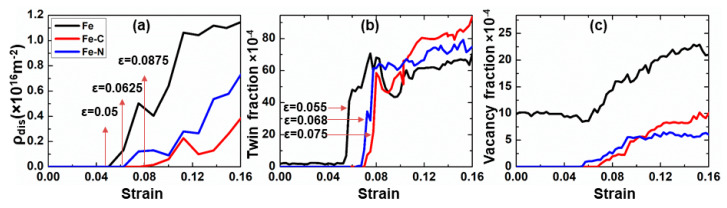
Evolution of (**a**) intragranular dislocation density, $\rho_{dis}$, (**b**) twin fraction, and (**c**) vacancy fractions with applied strain for different (nc) samples.

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
