# Peer review of "Deformation Behavior of Nanocrystalline Body-Centered Cubic Iron with Segregated, Foreign Interstitial: A Molecular Dynamics Study"

_materials, 2020, doi:10.3390/ma13235351_

Round 1
Reviewer 1 Report
This manuscript reports the effects of carbon and nitrogen on the stress or strain of the material. The authors state that their results align with the results obtained from the Density Function Theory (DFT) where it is reported that adding carbon and nitrogen improves the cohesion of GB in bcc-Fe. The study is important but it lacks several important information. The paper is relatively well written but the improvement text is relatively clear. However, it is not clear what is the new result here compared to the results already published earlier. In addition, the abstract and conclusion sections do not provide any concrete results. I encourage authors to revise both the abstract and conclusion section and support their claim with the data and numbers. For example, the content of C or Ni in % in the alloy. And, strain or stress in %? What do the authors mean by low or high strain or stress and Fe-based alloys? Authors should address all the above-mentioned concerns, they should use chemical formulae where applicable, provide quantitate information instead of qualitative alone. Authors should also improve the figure quality, for example, Figs. 3 & 8 (where is the guide for the eyes?).
Recommendation: A revision is needed prior to acceptance.
Reviewer 2 Report
Review for the manuscript #Materials974367
In the proposed manuscript, the authors present simulations of Molecular Dynamics (MD) to study the impact of C and N atoms on the deformation of a Fe-polycrystal with nanometer-sized grains. The paper is interesting, well written most of the time and corresponds perfectly to the interests of the journal's audience. However, I have many comments that I kindly suggest to the authors to take into account before publication.
- As usual with MD simulations, the choice of the semi-empirical potentials describing the interactions of Fe with impurities controls most of the simulation results and this is not discussed in this manuscript. This point is even more important for Fe bcc which is magnetic and the impurity elements found in the interstitial sites. The paper therefore lacks a long section dealing with the choice of the interatomic potentials. In particular, it should demonstrate the ability of these potentials to describe the compact core of the screws, some interaction energies between Fe with impurities and some surface energy values or Grain Boundary (GB) energy values. All this should be compared with reference data from experiments or DFT calculations.
- Some details regarding the simulation configurations are missing. Nothing is said regarding the number of grains and their sizes. It would have been interesting to provide some data regarding the type of GB that are formed in the simulation. Since the GB are very flat (some of them are even low angle GB with GB dislocations), it is expected that some of these GB are special coincidence GB, that may be known in the GB community.
- the reviewer tends to think that the document lacks statistical analysis and that several simulations should have been proposed when only one is presented. For example, why not compare several nanocrystals obtained with different seeds. The simulation does not seem to be extremely expensive in terms of computing resources compared to what MD can do (see the billion atom simulations from Livermore Nat. Lab.). This is also the case for the distribution of impurities in the nanocrystal or for the stacking fault energy calculations.
- The usage of the ratios of figure 3 and 8 is not convincing at all, as nothing is said regarding the absolute values of the energies (or atomic volume). The distribution of energies (or atomic volume) of GB atoms would be more instructive.
- Some conclusions on rather complex topics are drawn a bit too quickly, The authors should be more nuanced in their conclusions. The paper appears a bit superficial when this occurs.
-
- For example line 46-47: Wu et al. [21] reveal that carbon acts as cohesion enhancer in bcc-Fe grain boundary while nitrogen weakens GBs[22]” more details should be provided on this study and how it relates to the present calculations
- Lines 142-144: "Our findings are in line with the results of Density Function Theory (DFT) where it is reported that adding carbon and nitrogen improves the cohesion of GB in bcc-Fe [22,39].” More details regarding these DFT calculations should be provided.
- Line 260: “The increase of the twinning stress of ferrite when carbon is added to ferritic steel was revealed experimentally by Magee et al. [47].” More context and details should be provided regarding this study.
- "Adding carbon or nitrogen to ferrite also reduces the free atomic volume fraction and, consequently, the grain boundary sliding resistance increases.” The correlation between GB sliding and the free atomic volume has been sometimes observed in the litterature but there is no consensus to the reviewer's knowledge.
- Line 153-158 “The present results support the explanation given by Takaki [18] for the higher value of the Hall-Petch coefficient ky for carbon-added ferritic steel compared to that for nitrogen-added ferritic steels, following Cottrell’s model [40], in which it is assumed that segregated carbon and nitrogen pin the dislocation emission site at grain boundary." The Hall-Petch effect remains to this day poorly known with many impacting parameters such as crystalline orientation, deformation, grain size and shape temperature… This sentence should more nuanced.
- Some additional minor comments. The usage of ferrite in the title is misleading. The authors should check the usage of cohesive energy in the text. What is the reference energy used to calculate the GB excess energy in presence of impurities?
